# The Economic Policy Uncertainty and Its Effect on Sustainable Investment: A Panel ARDL Approach

**Susilo Nur Aji Cokro Darsono** [1,2,*], **Wing-Keung Wong** [3,4,5], **Tran Thai Ha Nguyen** [1,6] **and Dyah Titis Kusuma Wardani** [2]

[1] Department of Business Administration, College of Management, Asia University, Taichung City 41354, Taiwan; nguyen.tranthaiha@sgu.edu.vn
[2] Department of Economics, Faculty of Economics and Business, Universitas Muhammadiyah Yogyakarta, Yogyakarta 55183, Indonesia; dyah.wardani@umy.ac.id
[3] Department of Finance, Fintech & Blockchain Research Center, and Big Data Research Center, Asia University, Taichung City 41354, Taiwan; wong@asia.edu.tw
[4] Department of Medical Research, China Medical University Hospital, Taichung City 41354, Taiwan
[5] Department of Economics and Finance, The Hang Seng University of Hong Kong, Shatin 999077, Hong Kong
[6] Faculty of Finance and Accounting, Saigon University, Ho Chi Minh City 756100, Vietnam
* Correspondence: susilonuraji@umy.ac.id

**Abstract:** This study examines the effect of economic policy uncertainty (EPU) on sustainable investment returns by using panel data of stock market returns and the EPU index from twelve countries for the period from April 2015 to December 2020. In addition, precious metal prices, energy prices, and cryptocurrency prices are used as control variables. To do so, we investigate the impact of EPU, gold prices, oil prices, and Bitcoin prices on stock market returns by using the panel autoregressive distributed lag (ARDL) model to examine both the long-run correlation and short-run effect. Our findings show that EPU, gold prices, oil prices, and Bitcoin prices have a time-varying significant impact on sustainable stock market returns. We discovered that EPU has a significantly negative impact on the returns of the sustainable stocks in the markets over the long run. In contrast, the rise of the gold price, oil price, and Bitcoin price have a significantly positive impact on the returns of the sustainable stocks in the twelve sustainable markets in the long run. On the other hand, EPU in Singapore, Spain, the Netherlands, and Russia has a significant short-run impact on market returns in each country. Based on the findings, managers and investors in the sustainable stock markets are highly recommended to pay more attention to the volatility of EPU, gold prices, oil prices, and Bitcoin prices in the short run to control the risk of returns in the sustainable stock market. Furthermore, policymakers must closely monitor the movement of the EPU index, as it is a major driver of sustainable stock market returns.

**Keywords:** economic policy uncertainty; EPU; energy prices; panel ARDL; sustainable investment; sustainable stock market returns

## 1. Introduction

Policy changes are inevitable and might come unexpectedly. Uncertainty drives a range of adjustments in individual and organizational decisions. Either political, social, economic or both regulations are referred to catch recent occurrences or forecast the dynamic progress. Dynamism sometimes turns into uncertainty, which can cause shocks in the stock market. When the government announces many policy changes and leaves a significant amount of uncertainty, it will bring about volatilities and correlations among stocks (Pastor and Veronesi 2012). Baker et al. (2016) developed the Economic Policy Uncertainty (EPU) index by using several data as a proxy for uncertainty in economic policy movements and found that EPU yields significant impacts on both micro and macroeconomic factors. They also discovered that EPU had a considerable influence on the decline in both

firm-level investment and employment, as well as the economy as a whole. EPU was more substantial and impactful than other variables that hinder developing stock markets, such as geopolitical risk and financial stress (Das et al. 2019).

In several prior types of research, the influence of economic policy uncertainty on stock market returns has been explored. Since the commencement of the global financial crisis in 2008 and the stock market fall in China in 2015, interest in this impact has reemerged (Nakai et al. 2016; Yuan et al. 2022b). One decade ago, Pastor and Veronesi (2012) created a model of general equilibrium that predicts a decline in stock values after the announcement of a change in government policy. In light of increased policy uncertainty, the probability of a decrease in stock prices will increase. The authors then extend the basic model to demonstrate that political instability imposes a bigger risk premium in weaker economic circumstances. Further, Antonakakis et al. (2013) investigate the effect of policy uncertainty, implied volatility, and stock market returns in time-varying circumstances. They employ the S&P500 returns, the VIX, and the economic policy uncertainty (EPU) index of Baker et al. (2016). The author discovered that the dynamic linkage between EPU and stock market returns was persistently negative over time and sensitive to shocks in oil demand and US recessions. On a similar note, Sum (2013) revealed that the United States EPU Index is negatively linked to the returns of five ASEAN stock markets.

In addition to the literature on the effect of EPU on the relationship between financial markets, Yu and Huang (2021) employed the GARCH-MIDAS model in their latest empirical investigations. They discovered that the fluctuating uncertainties of Chinese economic policy might contribute to the instability of the Chinese stock market. Further, Kundu and Paul (2022) examined the effect of economic policy uncertainty on stock market returns and risk for G7 nations and found that EPU had a negative influence on stock market returns. Increases in EPU were observed to increase market volatility and decrease returns during contemporaneous periods. The vast majority of current empirical research has shown that policy uncertainty's impact on stock market returns is negative, while the influence on volatility is positive. From the empirical evidence, volatilities arise (Dakhlaoui and Aloui 2016; Su et al. 2018), and the stock market returns decline significantly (Arouri et al. 2016; Christou et al. 2017; Guo et al. 2018; Hashmi et al. 2021; Kang et al. 2017).

Stock market reactions to economic policy uncertainty do not apply uniformly to all nations. A general prediction that the announcement of policy changes would result in a stock market decline is not always justified (Chang et al. 2015). Wu et al. (2016) explained that the final result of the causal relationship between EPU and the stock market may vary among nine selected countries. EPU may only have a significant impact on stock prices when it exceeds a certain level. In contrast, its effect might differ considerably based on whether the stock markets are in bullish or bearish phases. Yang and Jiang (2016) revealed that policy uncertainty and stock returns only generated a weak dynamic correlation coefficient. This result indicated that their values mainly influenced the fluctuations of each variable in the preceding period. In the study for G7 countries, Raza et al. (2018) discovered that EPU had varying effects on the equity premium across countries. This heterogeneity exists due to the dependence of each nation on economic policies, other stock markets, and diverse nations. Most of the existing literature focuses on the impact of EPU on stock market returns, volatility, liquidity, economic growth, inflation, and firm investments (see, among others, K. L. Chang 2021; Chiang 2019; Dakhlaoui and Aloui 2016; Liu and Zhang 2015; Luo and Zhang 2020; Sum 2013; Vo et al. 2021; Wang et al. 2022; Wu et al. 2016; Xu et al. 2021).

During the last several years, the link between EPU, gold prices, and oil prices has attracted considerable attention with respect to its impact on the stock market. In a more recent body of research, Zhao and Wang (2021) found that the US stock market is more closely related to the crude oil and gold markets than the Chinese stock market. The positive effect was detected in the correlations between crude oil and stocks. However, EPU had a negative influence on the correlations between gold and stocks. Another strand

of evidence demonstrated a favorable effect of EPU and long-term oil-stock correlations (see Fang et al. 2018; Prawoto and Putra 2020; Oliyide et al. 2021; Yang et al. 2021).

In this age of disruption, cryptocurrencies like Bitcoin and Ethereum are becoming a high-return investment opportunity. Indeed, the high return associated with crypto invest-ments is accompanied by substantial risks. Consequently, the emergence of cryptocurren-cies may affect policy uncertainty and the stock market. Extending the literature to the effect of EPU on the relationship between financial markets and Bitcoin, Matkovskyy et al. (2020) concluded that the interdependence between traditional financial markets and Bitcoin decreases due to economic uncertainty shocks. The authors discovered that uncertainty shocks in US economic policy are connected with a reduction in volatility in the Bitcoin markets that were investigated. In addition, a rise in economic uncertainty in Japan reduces the volatility of the JPY Bitcoin market. Further, Fasanya et al. (2021) confirmed that the connection between Bitcoin and precious metals might not act as a hedge or safe haven against economic policy uncertainties around the research periods. In contrast, research conducted by Hussain Shahzad et al. (2020) in G7 nations revealed that Bitcoin serves as a distinct safe haven and hedge for Canadian stock indexes. The increase in the EPU level was related to a rise in the ideal Bitcoin portfolio weight prior to the 2007 Bitcoin crash. The EPU had a detrimental impact on the dynamic conditional correlations between Bitcoin and the US stock market following the Bitcoin crash (Ahmed 2021; Mokni et al. 2020).

Sustainable investment has attracted the attention of both academics and investors over the last several decades due to its rapid development. It is also known as ethical investment, which is also referred to as green investment, ESG investment, or socially responsible investment (SRI) (Escrig-Olmedo et al. 2017). The term "sustainable" describes innovative thinking that aims to bring about genuine change by integrating climate, economics, and ethical considerations. The triple bottom line (TBL) concept considers environmental, economic, and social aspects. Sustainable investment becomes more effective when TBL aspects are taken into account (Tseng et al. 2019). However, research on the effects of EPU, commodities, and cryptocurrencies on sustainable investing (such as a green bond, green fund, green Sukuk, SRI fund, Sustainable stock market or ESG In-vestment, impact investment, and value investment) is still limited. Thus, in this paper, we bridge the gap of the literature to investigate the issue.

Pham and Nguyen (2022) examined the linkages between the green bond market, financial uncertainty, and economic uncertainty from 2014 to 2020. The study used four major green bond indices in the United States and Europe with OVX as oil uncertainty, VIX as stock uncertainty and the US EPU index. The author discovered that the spillover effects from financial and economic uncertainty on the green bond market are smaller but more persistent in the low-uncertainty state. Nevertheless, the spillover effects are larger but less persistent in the high-uncertainty state. As the number and scope of SRI values have risen, academics, practitioners, and policymakers have been compelled to weigh the pros and cons of making non-traditional stock market investments by examining the interactions between a company's social, environmental, and financial performance (Garcia et al. 2017). At the same time, Escrig-Olmedo et al. (2017) concluded that business, environmental, financial, and social governance are the key factors in making sustainable investment decisions for institutional investors. Further, Darsono et al. (2022) examined the effect of good governance on sustainable investment in the Asian region. They discovered that political stability and regulatory quality are linked to higher sustainable investment returns. This study investigated the relationship between economic policy uncertainty (EPU) and sustainable investment, as measured by the returns of the sustainable markets of twelve countries from 2015 to 2020. The volatility of commodity prices and the rise of cryptocurrency as an alternative investment may impact stock returns. Therefore, we add the prices of gold, oil, and Bitcoin as control variables. The Panel ARDL model with Mean Group (MG) and Pooled Mean Group (PMG) estimations by Pesaran et al. (1999) is utilized to examine the short-run and long-run effect of EPU, gold prices, oil prices, and Bitcoin prices on the returns of the sustainable markets.

This study contributes to the current body of knowledge in several ways. First, this paper examines the short-run and long-run correlation between economic policy uncertainty and sustainable stock market returns using panel data under time-varying and diverse market conditions. Second, this paper examines the short-run effect of EPU, global oil, gold, and Bitcoin prices on sustainable stock market returns for individual nations using PMG estimation. The findings could enrich the literature and provide beneficial insights for policymakers, investors, and investment managers in monitoring and developing sustainable investments.

## 2. Data and Methods

### 2.1. Data

The data series used in this study include sustainable stock market returns, economic policy uncertainty (EPU), oil prices, gold prices, and Bitcoin prices while the sample of the returns data for the sustainable stock markets is screened based on the Sustainable Stock Exchanges Initiatives by United Nations (SSE Initiatives 2010). Due to the availability of data related to nations with both an EPU index and a sustainable stock market index obtained from Thomson Reuters DataStream, this study only uses data of monthly returns from April 2015 to December 2020 for the sustainable stock market and EPU from 12 countries in our analysis, including China (SSECGI), India (NIFTY100 ESG), Japan (JPXNK400), Singapore (iEdge SG ESG), Germany (SXWESGU), Netherlands (EURONEXT100), Russia (MRSV), Spain (IBEXFG), UK (FT4GDBUK), Brazil (ISE B3), Colombia (COLIR), and the US (FT4GDBUS). Despite higher frequency data being available for stock returns, in this paper we only use the monthly return of the stock prices because EPU can be obtained only for monthly basis. The monthly return of the stock market is computed as $SSMR_{it} = [\log(P_{it}) - (\log(P_{it-1})]$, where $P_{it}$ is the monthly closed price for the ith country at the tth month.

The economic policy uncertainty (EPU) index found at http://www.policyuncertainty.com accessed on 1 February 2021 and obtained from the Economic Policy Uncertainty database created by Scott R Baker, Nick Bloom, and Steven J. Davis (Baker et al. 2016) is based on the frequency of the coverage in ten leading newspapers. The digital archives of each newspaper were searched for the monthly number of articles by using a particular string of keywords. For example, the articles must include three words: "uncertain" or "uncertainty"; "economy" or "economic"; in addition to any of the following policy terms: "deficit", "Congress", "Federal Reserve", "Regulation", "Legislation", and "White House". In short, the EPU index refers to economic policy problems and anticipated (or actual) changes in government policy and related issues (Baker et al. 2016).

In this study, we include three control variables consisting of the global prices of two commodities, such as gold and oil, and a cryptocurrency as the current investment option, such as the price of Bitcoin.

### 2.2. Research Method

The research method consists of several steps. The first step before testing the variables is to check whether there is an outlier or missing data in the data of both the economic policy uncertainty and stock prices. Thereafter, the data will be analyzed by using the Panel autoregressive distributed lag (ARDL) models with two estimations: Mean Group (MG) and Pooled Mean Group (PMG) estimations. The panel ARDL model was then applied to examine both the short-run and long-run correlations between EPU, and the returns of gold prices, oil prices, Bitcoin prices, and sustainable stock markets.

#### 2.2.1. Panel Unit Root Tests and Cointegration Tests

We first apply the unit root tests the applied time series before conducting the cointegration analysis in our analysis, including the most relevant panel unit root tests developed by Im et al. (2003) and Levin et al. (2002) to establish the order of integration of each variable. These tests are distinguished by the presence of a null hypothesis in which all the panels have a unit root. We run each test with variables in both levels and initial differences.

After determining the panel unit root, we use the traditional panel cointegration tests developed by Kao (1999) and Pedroni (1999, 2004) to check whether it is possible to construct a long-run equilibrium relationship between these variables being studied in our paper. Traditional panel cointegration tests such as Kao (1999) & Pedroni (1999, 2004) might be used. However, if both variables do not coincide with the same order of integration, we use another strategy as discussed later.

### 2.2.2. Panel Model ARDL

The Panel Autoregressive Distributed Lag (ARDL) model will be used if no cointegration is identified by applying the preceding methods. We use this model in our study because it is superior regardless of whether the underlying regressors exhibit I(0), I(1), or a combination of both (Pesaran and Shin 1999). The macro panel data method can be implemented across a period of more than 20 years. Due to the nature of the dataset, it was not appropriate to utilize the GMM estimator. Following the vast literature on dynamic panel data, we use both Mean Group (MG) and Pooled Mean Group (PMG) estimators to analyze the link between economic policy uncertainty and other variables and study their impacts on sustainable investment (Pesaran et al. 1999; Pesaran and Smith 1995). The choice of a pooled regression enhances the number of observations (degrees of freedom), which are limited in macroeconomic studies due to the lower frequency of available observations. This improves the accuracy of estimation. The ARDL approach is used due to its flexibility in controlling variables with different degrees of integration. The main model of the panel ARDL approach is:

$$SSMR_{it} = \alpha_i + \sum_{l=1}^{p} \beta_0 SSMR_{i,t-1} + \sum_{l=0}^{q} \beta_1 EPU_{i,t-1} + \sum_{l=0}^{q} \beta_2 x_{i,t-1} + u_{it}. \tag{1}$$

By reparameterising Equation (1), we have:

$$\Delta SSMR_{it} = \alpha_i + \Phi_i (SSMR_{i,t-1} - \theta_1 EPU_{i,t-1} - \theta_2 x_{i,t-1}) + \sum_{l=1}^{p-1} \lambda_{il} \Delta SSMR_{i,t-1} +$$
$$\sum_{l=0}^{q} \lambda'_{il} \Delta EPU_{i,t-1} + \sum_{l=0}^{q} \lambda''_{il} \Delta x_{i,t-1} + u_{it}, \tag{2}$$

where $i$ and $t$ represent country and time, respectively, *SSMR* represents the sustainable stock market returns, *EPU* denotes the economic policy uncertainty, and $x$ represents a set of control variables including: gold price, oil price, and Bitcoin price. In notation, the short-run coefficients of the lagged dependent variable and other control variables are $\lambda$, $\lambda'$, $\lambda''$, respectively; the long-run coefficients in our model are $\theta_1$ and $\theta_2$; and $\Phi_i$ is the speed of adjustment.

The Pesaran et al. (1997) PMG considers long-term slope parameters to be homogenous across countries but short-run coefficients to be heterogeneous. MG provides for country-specificity in both the short and long term. This technique calculates unique regressions for each country before computing unweighted means. To distinguish between the PMG and MG, we use the Hausman test to see if there are any significant differences between these estimators. Although PMG and MG are both consistent, PMG is more efficient under the premise of long-term homogeneity (Pesaran et al. 1999).

### 3. Empirical Analysis

This study used monthly panel data from the sustainability index and EPU index in twelve countries from April 2015 to December 2020. The following summarizes the price movement of the sustainable stock market price in twelve countries during the research period, which was combined with the individual EPU index through a time series plot.

Figure 1 shows that the movement pattern of EPU and stock prices had correlations for each country. The EPU movement in 2016 is relatively high for some countries such as India, Japan, Singapore, Germany, the UK, Brazil, and the US. The movement of the stock price up in the long term is found in most of the countries except in China, Singapore,

Spain, the UK, and Colombia which turn down in 2020. This downturn might be affected by the pandemic of COVID-19 that caused uncertainty in policy and a slowdown of the global economy.

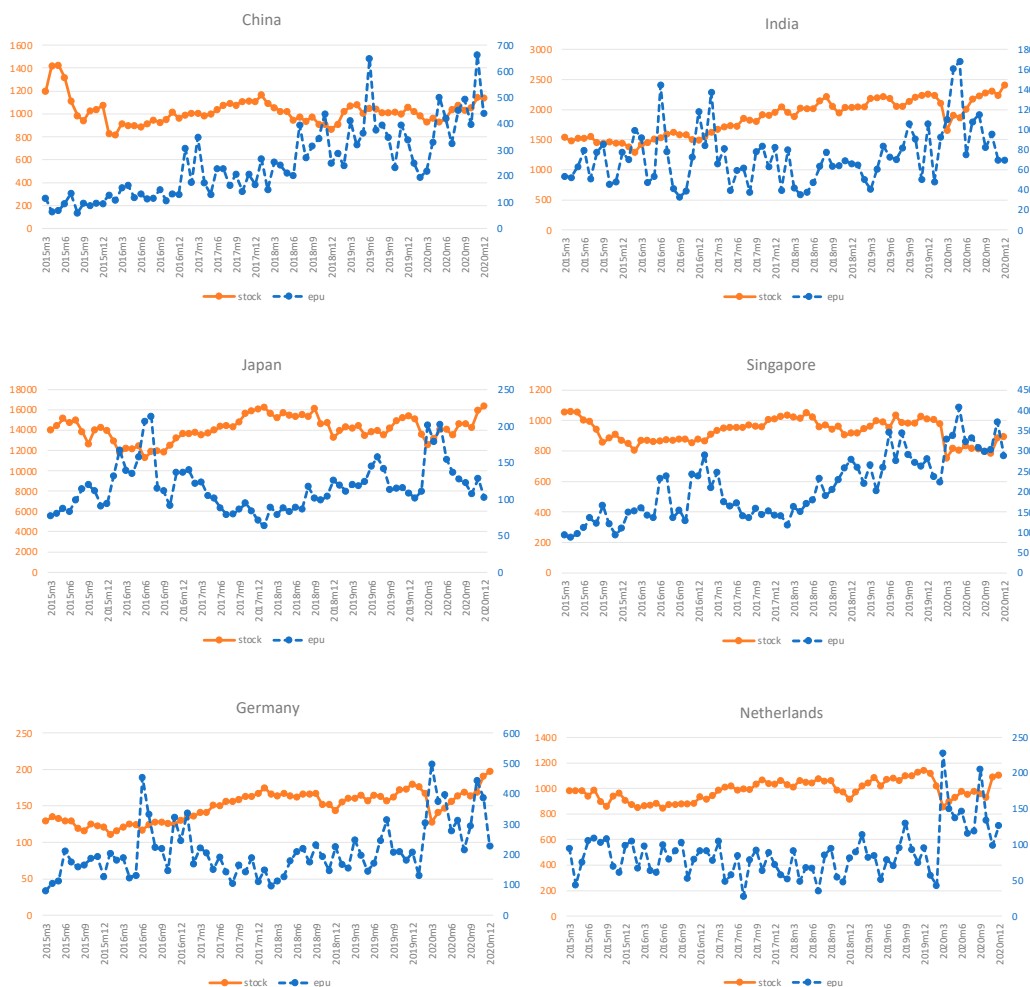

**Figure 1.** *Cont.*

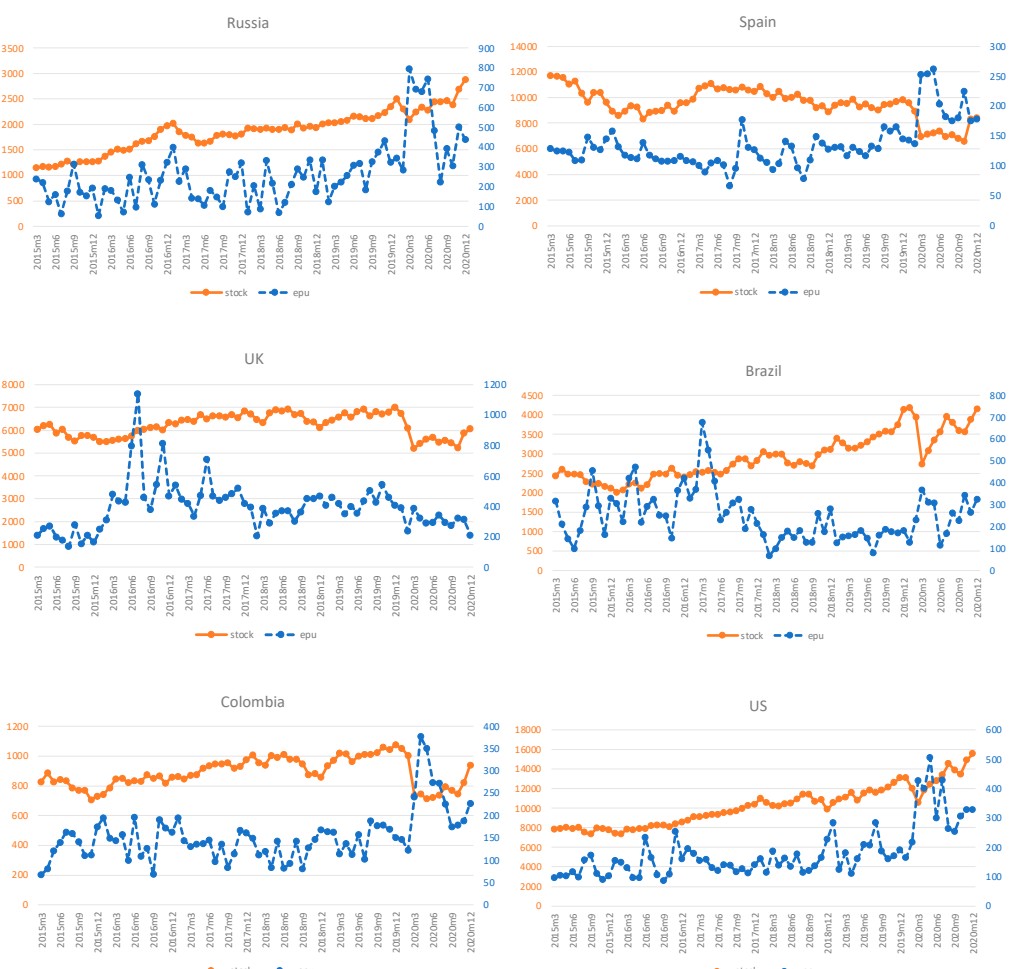

**Figure 1.** The movement of sustainable stock market price and EPU for 12 countries.

### 3.1. Descriptive Statistics

Table 1 presents the summary statistics of the average monthly sustainable stock market returns for twelve countries. The monthly average return of the sustainable stock market is positive for all countries. All the stock return series are found to be negatively skewed except China, Brazil, and the US, and platykurtic except for China and Spain.

**Table 1.** Summary statistics of Sustainable Stock Market Index in 12 Countries.

| Countries | Index | Mean | Max | Min | Std. Dev. | Skew. | Kurt. |
|---|---|---|---|---|---|---|---|
| China | SSECGI | 1021.13 | 1425.45 | 816.40 | 110.77 | 1.47 | 3.98 |
| India | NIFTY100 ESG | 1852.85 | 2409.95 | 1286.13 | 300.57 | −0.09 | −1.35 |
| Japan | JPXNK400 | 14,134.92 | 16,341.80 | 11,250.20 | 1205.22 | −0.30 | −0.38 |
| Singapore | iEdge SG ESG | 930.79 | 1059.65 | 753.60 | 76.53 | −0.26 | −0.85 |
| Germany | SXWESGU | 149.07 | 197.00 | 111.00 | 20.40 | −0.06 | −0.95 |
| Netherlands | EURONEXT100 | 985.54 | 1144.39 | 846.47 | 79.54 | −0.04 | −1.01 |
| Russia | MRSV | 1865.22 | 2875.55 | 1152.78 | 397.46 | −0.01 | −0.32 |
| Spain | IBEXFG | 9503.16 | 11,690.40 | 6576.10 | 1203.78 | −0.62 | 0.20 |
| UK | FT4GDBUK | 6231.08 | 7024.85 | 5211.55 | 494.37 | −0.31 | −1.11 |
| Brazil | ISE B3 | 2902.70 | 4182.17 | 2011.55 | 570.74 | 0.57 | −0.57 |
| Colombia | COLIR | 889.60 | 1075.67 | 704.79 | 99.50 | −0.07 | −1.05 |
| US | FT4GDBUS | 10,203.73 | 15,556.21 | 7338.26 | 2050.17 | 0.51 | −0.42 |
| Total Obs | | | | 840 | | | |

Note: The sample period is from 2015M03 to 2020M12.

The summary statistics of economic policy uncertainty (EPU) for twelve countries are presented in Table 2. The variability of EPU is observed to be greatest in the United Kingdom. UK EPU has a maximum value of 1141.8 and a standard deviation of 159.67. The Netherlands has the lowest EPU fluctuation, with 27.63 and a standard deviation of 34.65. All of the EPU indices in the data are positively skewed and leptokurtic.

**Table 2.** Summary statistics of economic policy uncertainty index.

| Countries | Mean | Max | Min | Std. Dev. | Skew. | Kurt. |
|---|---|---|---|---|---|---|
| China | 247.46 | 661.83 | 60.21 | 137.45 | 0.88 | 0.50 |
| India | 73.55 | 167.75 | 32.88 | 28.73 | 1.17 | 1.76 |
| Japan | 116.34 | 212.49 | 63.29 | 32.57 | 1.14 | 1.36 |
| Singapore | 210.18 | 406.95 | 86.58 | 78.76 | 0.39 | −0.80 |
| Germany | 211.75 | 498.06 | 80.56 | 88.87 | 1.28 | 1.50 |
| Netherlands | 88.15 | 228.70 | 27.63 | 34.65 | 1.50 | 4.26 |
| Russia | 261.30 | 793.63 | 55.10 | 155.61 | 1.51 | 2.85 |
| Spain | 134.48 | 261.61 | 66.65 | 38.35 | 1.53 | 2.79 |
| UK | 393.69 | 1141.80 | 137.50 | 159.67 | 1.84 | 6.73 |
| Brazil | 247.27 | 676.96 | 68.46 | 113.96 | 1.16 | 2.06 |
| Colombia | 153.29 | 376.84 | 67.82 | 56.57 | 1.64 | 4.39 |
| US | 180.22 | 503.96 | 86.34 | 88.09 | 1.72 | 2.95 |
| Total Obs | | | | 840 | | |

Note: The sample period is from 2015M03 to 2020M12.

Table 3 presents the summary statistics of control variables such as gold, oil, and Bitcoin global prices. It can be seen that Bitcoin was growing rapidly from 229 USD for 1 to a maximum reach of 5483.87 USD for 1 BTC. While the maximum oil price was 77 USD per barrel and the maximum gold price was 1969 USD per troy ounce during this research period.

**Table 3.** Summary statistics of GOLD OIL Bitcoin prices (in US Dollar).

| Summary | GOLD | OIL | BITCOIN |
|---|---|---|---|
| Mean | 1352.56 | 53.47 | 5483.87 |
| Max | 1969 | 77 | 28933 |
| Min | 1068 | 30 | 229 |
| Std. Dev | 217.25 | 11.03 | 5256.25 |
| Skew | 1.37 | 0.05 | 1.51 |
| Kurt | 1.03 | −0.60 | 4.09 |

Note: The sample used global prices in US dollars.

All variables were converted into the natural log for further analysis in order to provide comparable units of measurement and lessen the skewness of the original data.

*3.2. Panel Unit Root Test and Cointegration Test*

The first test applied for the panel ARDL approach is the panel unit root test. The unit root tests that are applied for the panel data in this research are Levin-Lin-Chiu (LLC) and Im-Pesaran-Shin (IPS) (Im et al. 2003; Levin et al. 2002). The result given in Table 4 shows that all variables are I(1) and I(0) in log level and log first difference, respectively. The result of LLC and IPS of EPU is stationary I(0) while sustainable stock market returns (SSMR), gold price (GOLD), oil price (OIL), and Bitcoin price (BITCOIN) are integrated of I(1) in the IPS test. None of the variables is integrated of an order greater than 1, indicating the appropriateness of the Panel ARDL approach. This level of integration might have been affected by structural breaks such as the Chinese stock market crash of 2015 and the Pandemic COVID-19 crisis of 2020 (Boateng et al. 2021; Broadstock et al. 2021; Chiah and Zhong 2020; Jin et al. 2019; Li et al. 2017; Luo and Zhang 2020).

**Table 4.** Panel Unit Root Test Results.

| Variable | Levin-Lin-Chiu | | Im-Pesaran-Shin | |
|---|---|---|---|---|
| | Level | 1st Difference | Level | 1st Difference |
| SSMR | −4.034 *** | −32.645 *** | −3.327 | −31.051 *** |
| EPU | −9.457 *** | | −9.124 *** | |
| GOLD | 1.201 | −26.593 *** | 3.277 | −25.469 *** |
| OIL | −3.172 | −43.928 *** | −0.7404 | −40.965 *** |
| BTC | −4.486 *** | −28.3066 *** | −0.9973 | −26.481 *** |

Note: *** denotes 1% significance level.

This study has two cointegration tests, including the Kao Test and the Pedroni Test. The results of both test results are reported in Table 5. These show that all tests support a cointegration relation given that the null hypothesis is rejected.

**Table 5.** Panel Cointegration Test Results.

| | Kao Test | | | Pedroni Test | |
|---|---|---|---|---|---|
| | Statistics | *p*-Value | | Statistics | *p*-Value |
| Modified Dickey-Fuller | 3.138 | 0.0008 | Modified Phillips-Perron | −6.385 | 0.000 |
| Dickey-Fuller | 4.373 | 0.000 | Phillips-Perron | −9.006 | 0.000 |
| Augmented Dicky-Fuller | 5.107 | 0.000 | ADF | −9.286 | 0.000 |
| Unadjusted modified DF | −7.846 | 0.000 | | | |
| Unadjusted DF | −4.0002 | 0.000 | | | |

*3.3. Panel ARDL Approach*

This section presents the results of the dynamic panel ARDL long-run and short-run effects of economic policy uncertainty (EPU), gold price (GOLD), oil price (OIL), and Bitcoin price (BITCOIN) on sustainable stock market returns (SSMR) by considering the Mean Group (MG) and Pool Mean Group (PMG) estimations. In addition, we applied the Hausman test to clarify which is the best method to achieve consistency and efficiency.

Based on Table 6, it can be observed that in the long-run PMG estimation, EPU negatively affected sustainable stock market returns (SSMR) at a 5% significance level, while gold price (GOLD), oil price (OIL), and Bitcoin (BTC) had a positive impact on sustainable stock market returns at a 1% significance level.

By contrast, in the short-run MG and PMG estimation, EPU also had a negative effect and GOLD had a positive effect, but they were not significant, while oil price (OIL) and Bitcoin price (BTC) positively impacted SSMR at a 1% significance level.

Based on the result of the Hausman test, we cannot reject the null hypothesis of the homogeneity restriction related to the sustainable stock market returns in the long term and short term. This highlights that the PMG is more efficient than the MG; hence, we focus our interpretation on the PMG estimator. Henceforth, according to the PMG, there is a negative relationship between economic policy uncertainty in the long run with a coefficient of −0.071. It means that the increase of 1 percent of EPU will cause a decrease of 0.071% in sustainable stock market returns.

The prices of commodities also had a positive and significant impact in the long run, such as gold price with a coefficient of 0.503, oil price coefficient of 0.254, and Bitcoin price with a coefficient of 0.047.

**Table 6.** MG and PMG estimation results.

| | Estimator | MG | | PMG | |
|---|---|---|---|---|---|
| | Variable | Coefficient | *p*-Value | Coefficient | *p*-Value |
| Long Run | EPU | −0.052 * (0.0226) | 0.022 | −0.071 ** (0.025) | 0.004 |
| | GOLD | 0.256 * (0.119) | 0.031 | 0.503 *** (0.096) | 0.000 |
| | OIL | 0.204 *** (0.029) | 0.000 | 0.254 *** (0.053) | 0.000 |
| | BTC | 0.0323 *** (0.009) | 0.000 | 0.047 *** (0.009) | 0.000 |
| Short Run | Ect | −0.490 *** (0.032) | 0.000 | −0.174 *** (0.044) | 0.000 |
| | EPU | −0.010 (0.009) | 0.242 | −0.019 (0.013) | 0.137 |
| | GOLD | 0.0259 (0.045) | 0.572 | 0.061 (0.069) | 0.378 |
| | OIL | 0.067 *** (0.016) | 0.000 | 0.116 *** (0.020) | 0.000 |
| | BTC | 0.0459 *** (0.005) | 0.000 | 0.048 *** (0.006) | 0.000 |
| | Cons | 2.572 (0.587) | 0.000 | 0.433 (0.110) | 0.000 |
| Hausman Test | | | | 6.16 (0.8015) | |

Note: those in ( ) are standard errors; *, **, and *** denote significance at 10%, 5%, and 1% levels.

### 3.4. PMG Individual Nations Short-Run Results

Individual countries' short-run coefficients estimated using the PMG estimator are shown in Table 7, in which the negative and significant ECT coefficients represent an adjustment speed for the long-run equilibrium relationship. There are three countries with a negative and significant relationship between EPU and Sustainable Stock Market Returns: Singapore, Spain, and the Netherlands, while Russian EPU had a positive and significant effect on Russian sustainable stock market return in the short run.

Then, the short-run relationship between gold price (GOLD) and sustainable stock market returns is positive and significant in four countries such as China, Brazil, the US, and India, while in the UK, the effect of gold price and stock market returns is negative and significant. This paper also considers the effect of oil price on sustainable stock market returns. The results show that oil price had a positive and significant influence on stock market returns in eight countries with sustainability indices in the short run. Further, Bitcoin was also tested to examine the relationship with the sustainable stock market returns in the short run. The result shows that Bitcoin price had a positive and significant effect on stock market returns in all countries, except Brazil, which is not significant.

Further, the results from Table 7 can be inferred that a significant long-run relationship was found between the variables for eight nations as evidenced by negative and significant error correction terms. Five nations had a significance level of 1%: India, Japan, Germany, Russia, and Brazil. While the Netherlands is significant at a 5% significance level, China and Colombia are significant at a 10% significance level.

**Table 7.** PMG individual nation results.

| Nation | ECT | D(EPU) | D(GOLD) | D(OIL) | D(BTC) | Const |
|---|---|---|---|---|---|---|
| China | −0.093 * | 0.0136 | 0.448 *** | 0.130 *** | 0.05 *** | 0.208 |
| | (0.053) | (0.0183) | (0.140) | (0.042) | (0.014) | (0.143) |
| India | −0.374 *** | −0.00003 | 0.214 ** | 0.0512 | 0.06 *** | 1.05 *** |
| | (0.094) | (0.013) | (0.104) | (0.04) | (0.010) | (0.400) |
| Japan | −0.144 *** | −0.0306 | −0.185 | 0.0116 | 0.08 *** | 0.705 ** |
| | (0.051) | (0.034) | (0113) | (0.038) | (0.012) | (0.280) |
| Singapore | −0.055 | −0.066 *** | −0.113 | 0.193 *** | 0.019 * | 0.121 |
| | (0.046) | (0.022) | (0.115) | (0.037) | (0.012) | (0.113) |
| Germany | −0.494 *** | −0.020 | −0.013 | 0.040 | 0.076 *** | 0.182 |
| | (0.104) | (0.013) | (0.103) | (0.036) | (0.008) | (0.392) |
| Netherlands | −0.118 ** | −0.019 * | −0.063 | 0.089 ** | 0.054 *** | 0.259 |
| | (0.057) | (0.012) | (0.106) | (0.036) | (0.011) | (0.158) |
| Russia | −0.162 *** | 0.028 * | 0.133 | 0.067 | 0.038 ** | 0.476 ** |
| | (0.063) | (0.015) | (0.170) | (0.055) | (0.018) | (0.219) |
| Spain | −0.039 | −0.137 *** | −0.0213 | 0.188 *** | 0.036 ** | 0.170 |
| | (0.036) | (0.041) | (0.148) | (0.049) | (0.015) | (0.166) |
| UK | −0.028 | 0.006 | −0.171 * | 0.171 *** | 0.035 *** | 0.116 |
| | (0.036) | (0.016) | (0.095) | (0.030) | (0.010) | (0.155) |
| Brazil | −0.378 *** | 0.008 | 0.437 *** | 0.143 ** | 0.020 | 1.264 *** |
| | (0.093) | (0.0198) | (0.162) | (0.058) | (0.016) | (0.402) |
| Colombia | −0.112 * | −0.016 | −0.047 | 0.232 *** | 0.04 *** | 0.239 |
| | (0.064) | (0.020) | (0.144) | (0.048) | (0.015) | (0.164) |
| US | −0.087 | 0.0026 | 0.306 *** | 0.070 ** | 0.072 *** | 0.400 |
| | (0.057) | (0.015) | (0.103) | (0.036) | (0.010) | (0.267) |

Note: those in ( ) are standard errors; *, **, and *** denote significance at 10%, 5%, and 1% levels.

## 4. Discussion

### 4.1. Correlation between the EPU and Sustainable Stock Market Returns

According to the panel model of pooled mean group estimation results, we discover that the EPU index has a negative and significant impact on sustainable stock market returns in the long term. However, in the short term, EPU does not significantly affect the sustainable stock market returns. This indicates that an increasing economic policy uncertainty index in the long run might lead to a decrease in sustainable stock market returns, while in the short run the EPU index has no significant implication for asset pricing factors for the sustainable stock market in these twelve countries. This result extends the findings of Chiang (2019), Dai et al. (2021), Das et al. (2019), Luo and Zhang (2020), Xu et al. (2021), Yang et al. (2021), and Yuan et al. (2022a) with the analysis of the effect of EPU on stock market returns.

Next, we examine the effect of gold, oil, and Bitcoin prices on sustainable stock market returns. Based on the PMG estimation results in the long term, we found that gold price, oil price, and Bitcoin price positively and significantly affect the sustainability index's market returns. It can be implied that increasing gold, oil, and Bitcoin prices might lead to an increase in sustainable stock market returns in the long term. This finding supports Hussain Shahzad et al. (2020) who stated that gold and Bitcoin are the safe-haven components for stock indices. Including gold in sustainable investment also might lower the total risk (Robiyanto et al. 2021), while in the short term, the gold price has no significant effect on the stock returns. This may be because the volatility of gold prices in the short term is low. However, the shocks of oil and Bitcoin prices in the short term have a positive implication on the returns of the sustainability index.

### 4.2. Effect of EPU, Gold, Oil, and Bitcoin Prices on Individual Sustainable Stock Market Returns

In this study, we also examine the individual effect of countries' EPU indices on each sustainable stock market return in the short term. According to the PMG results, we found that the EPU negatively affected the sustainable stock market returns in three countries, such as Singapore, Spain, and the Netherlands. This indicates that increasing the EPU index

in each country will cause a decrease in sustainable stock market returns in these three developed countries. In contrast, the Russia EPU index positively and significantly affects Russia's sustainable stock market returns. It indicates that the increasing uncertainty policy will lead to an increase in stock market returns. From these findings, we can imply that the sustainable stock markets in Singapore, Spain, Netherlands, and Russia are sensitive to economic policy uncertainty shocks in the short term. This is in line with the results obtained by Antonakakis et al. (2013), Guo et al. (2018), and Kundu and Paul (2022) who found a negative and significant relationship between EPU and stock market returns in ASEAN and G7 Countries.

Then, we found that the relationship between gold price and sustainable stock market returns is positive and significant in four countries: China, Brazil, the US, and India. This indicates that increasing gold prices might increase sustainable stock market returns in these three emerging countries (China, Brazil, and India) and the US. In fact, these three countries—China, Brazil, and the US—are the biggest gold producers globally, and India is the second biggest consumer of gold in the world. This finding is related to (Hussain Shahzad et al. 2020) who found that gold is an effective hedge for stock indices in the US. This result also confirmed previous research findings that gold showed a significant relationship and had a role as a safe haven for the stock markets during uncertainty and high volatility (Mensi et al. 2017; Robiyanto et al. 2021; Zhao and Wang 2021). In contrast, in the UK, the effect of gold price and stock market returns is negative and significant. It implies that increasing of gold price might reduce the sustainable stock market returns in the UK. The linkage of gold and the stock market as a safe haven and most effective hedge in the UK (Hussain Shahzad et al. 2020) causes contradictory effects due to the fact that the UK is no longer a gold producer.

This paper also considers the effect of oil price on sustainable stock market returns in the short run. We found that oil price had a positive and significant influence on stock market returns in eight countries with a sustainability index. These eight countries include China, Singapore, Spain, the UK, Colombia, the Netherlands, Brazil, and the US. This indicates that increasing oil prices might increase the stock market returns in these eight countries. This finding is supported by the results from Kang et al. (2017), Pham and Nguyen (2022), Yuan et al. (2022a), and Zhao and Wang (2021), who found the connection between oil prices and stock market returns.

In addition, Bitcoin as a cryptocurrency asset was also tested to examine the relationship with sustainable stock market returns in the short run. The result shows that Bitcoin price had a positive and significant effect on stock market returns in all countries, except Brazil, which is not significant. This result indicates that, in the short run, increases in Bitcoin might lead to an increase in sustainable stock market returns in these eleven countries. This result confirmed the finding of Ahmed (2021) who stated that the volatility of Bitcoin tends to have positive and significant effects on stock market returns, especially in normal conditions.

*4.3. Practical Implications*

Based on our findings, this paper has several implications for policymakers, investment managers, and investors. First, since the sustainable stock market returns were negatively affected by EPU, the increase in the EPU level will result in getting lower returns in both the short run and long run. While investment managers and investors aim to get high profit and excess returns from their investment, they also care about the accompanied risk, because high returns are always accompanied by high risks. Thus, though the majority of investors want to get high returns, most rational and valuable investors also care about the risks. In order to mitigate risks, investment managers must regularly monitor a country's economic policy uncertainty before providing recommendations or drawing conclusions about the attainment of a sustainable investment portfolio. Investors may select a nation with a lower EPU level and have sustainable investment to allocate their wealth to maximize their profits. Furthermore, investment managers and investors must

consider the volatility of gold, oil, and Bitcoin prices in order to make high returns. Thus, in order to attract investors in sustainable investment, a firm listed in the sustainable stock market index must be able to mitigate risks, have excellent sustainability performance, and provide good returns for investors.

The results from our paper could be beneficial for policymakers in considering the influence of policy uncertainty on sustainable investment. Therefore, while formulating an economic policy, policymakers should also closely monitor the policy's spillover effects. When policymakers can minimize the level of uncertainty, volatility risks will be mitigated, and investment returns will be more reliable. Thus, it will encourage a greater number of investors to increase the cash flow toward sustainable investment activities.

## 5. Conclusions

This study examines both long- and short-run effects of economic policy uncertainty and commodity prices for commodities such as gold, oil, and Bitcoin as an alternative investment in the sustainable stock markets of 12 countries by using monthly data for the period from 2015 to 2020. We first apply the panel unit root tests developed by Levin-Lin-Chiu and Im-Pesaran-Shin (IPS) to this study. We then employ the cointegration tests in our analysis by using both the Kao test and Pedroni test to examine the integration of all of the variables. Further, we apply the dynamic panel autoregressive distributed lag (ARDL) technique to overcome the problem of different orders of integration among variables in our analysis. Another advantage of using this method is that it can distinguish the short and long-run relationships among variables. In particular, this study implements two alternative estimators: the Mean Group (MG) estimator and the Pooled Mean Group (PMG) estimator to examine the short-run and long-run effect of the EPU, gold prices, oil prices, and Bitcoin prices on the returns of the sustainable markets. Thereafter, the Hausman test was applied to find the most efficient and consistent estimator.

This paper first contributes to the existing literature on factors affecting sustainable investment, particularly in the sustainable stock market. We first investigated both the short- and long-term effects from the panel EPU, gold, oil, and Bitcoin prices on sustainable investment. We then discovered that EPU has a significantly negative effect in the long run, though it has no significant effect on sustainable stock market returns in the short run. Thereafter, we find that the prices of gold, oil, and Bitcoin have significantly positive impacts on the sustainable stock market returns in the long run, though they have no significant effect in the short run.

Second, this paper contributes to the existing literature by examining the effect of individual EPU on the return of individual sustainable stock market in the same period of study. Using the short-term PMG estimation, we discover that the sustainable stock markets from Singapore, Spain, and the Netherland are negatively and significantly affected by the individual country's EPU. We also find that the increase of Russia's EPU tends to increase Russia's sustainable stock market returns in the short run.

Third, this paper discovers that gold has a significantly positive impact on the returns of sustainable stocks in the China, Brazil, US, and India markets. In contrast, the gold price negatively affects the returns of the sustainable stocks in the UK market. On the other hand, oil is one of the highest demanded commodities that have positive impacts on the returns of the sustainable stocks in the markets in the eight countries, including China, Singapore, Spain, the UK, Colombia, the Netherlands, Brazil, and the US. In addition, we examine the effect of Bitcoin as one of the latest investment alternatives in the short run. By doing it, we confirm that Bitcoin has a positive and significant effect on the returns of the sustainable stocks in the markets of eleven countries, except Brazil.

This study has several limitations. For instance, in this study, we examine the influence on stock market returns without considering the risks and volatility. Also, the panel ARDL model used in this study is appropriate in the case of mixed variables; for example, some variables are stationary but others are nonstationary. However, this model restricts the variables under consideration to only one level of relationship and does not allow for a

greater number of long-run cointegrations. Due to the limited data available, the number of nations and observations covered in this research is restricted.

Thus, further study could examine the influence on stock market returns by considering both the risks and volatility, study the issue by using the model that could allow for a greater number of long-run relationships, and investigate the issue by using a dataset with longer period. Future research could also consider the potential effects of global EPU on various types of sustainable investment. This paper uses panel autoregressive distributed lag (ARDL) models to examine the effect of economic policy uncertainty, gold, oil, and Bitcoin prices on sustainable investment returns. Extensions of our paper could include using our approach to study other important issues, for example funding liquidity (Abbas et al. 2021), examining four-moment modified value at risk and conditional value at risk (using Cornish-Fisher Expansion) of different mixed portfolios pairing the assets under study (equity-oil and equity-gold portfolios) (Ali et al. 2021) and nearly non-stationary series (Cheng et al. 2021). There are many important issues to which academics and practitioners could apply the approach used in this paper. Readers may refer to Wong (2020) for more information.

**Author Contributions:** Conceptualization, S.N.A.C.D., T.T.H.N. and W.-K.W.; methodology, S.N.A.C.D., T.T.H.N. and W.-K.W.; software, T.T.H.N. and S.N.A.C.D.; validation, W.-K.W.; formal analysis, S.N.A.C.D.; resources, S.N.A.C.D., D.T.K.W.; data curation, S.N.A.C.D. and T.T.H.N.; writing—original draft preparation, S.N.A.C.D.; writing—review and editing, S.N.A.C.D., W.-K.W., T.T.H.N.; visualization, S.N.A.C.D. and T.T.H.N.; supervision, W.-K.W.; project administration, D.T.K.W.; funding acquisition, S.N.A.C.D., W.-K.W. and D.T.K.W. All authors have read and agreed to the published version of the manuscript.

**Funding:** This research has been supported by Asia University, Universitas Muhammadiyah Yogyakarta, China Medical University Hospital, The Hang Seng University of Hong Kong, Saigon University. Universitas Muhammadiyah Yogyakarta, grant number 550/PEN-LP3M/II/2020, Research Grants Council (RGC) of Hong Kong (project number 12500915), and the Ministry of Science and Technology (MOST, Project Numbers 106-2410-H-468-002 and 107-2410-H-468-002-MY3), Taiwan.

**Institutional Review Board Statement:** Not applicable.

**Informed Consent Statement:** Not applicable.

**Data Availability Statement:** In Data available in a publicly accessible repository that does not issue DOIs. Publicly available datasets were analyzed in this study. This data can be found here: [https://www.policyuncertainty.com/all_country_data.html] accessed on 1 February 2021.

**Acknowledgments:** The authors thank the Editor-in-Chief, Professor Thanasis Stengos, three anonymous referees, and the handling editor for their helpful comments which help to improve our manuscript significantly. The second author would like to thank Robert B. Miller and Howard E. Thompson for their continuous guidance and encouragement.

**Conflicts of Interest:** The authors declare no conflict of interest.

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
