# Peer review of "The Economic Policy Uncertainty and Its Effect on Sustainable Investment: A Panel ARDL Approach"

_jrfm, doi:10.3390/jrfm15060254_

Round 1
Reviewer 1 Report
The review report is attached in pdf, highlighted in colors when needed. Please refer to that.
Referee Report on
jrfm-1736548
“The Economic Policy Uncertainty and its Effect on Sustainable Investment: A Panel ARDL Approach”
submitted to “Journal of Risk and Financial Management (JRFM)”
May 2022
I have carefully reviewed the manuscript and my comments and concerns are available below.
The paper has numerous writing and logical errors throughout the manuscript, starting from the abstract.
For example,
Page 1, Lines 17-19: Following statements by the authors seem redundant in the abstract. “The quantitative research design will be used in this research. The data collection method uses secondary data obtained from the Thomson Reuters DataStream for stock data, economic policy uncertainty data from the EPU database and the World Bank database for macroeconomic data collection”. Suggestion: Please, improve the abstract. It is poorly written and not well organized.
Page 1, Lines 20-21: Authors’ statement “This study investigates the impact of EPU on stock market returns by using panel autoregressive distributed lag (ARDL) to examine the long-term cointegration and short-term effect”. Comment: Why only EPU? How about gold oil, and bitcoin? They are not judged using ARDL?
Page 1, Lines 22-23: Authors’ statement “The results of the autoregressive distributed lag regression indicate that EPU has a negative and significant effect in the long term. While gold, oil and bitcoin have a positive and significant impact on the sustainable stock market returns”. Comment: As the abbreviation “ADRL” has been used in earlier statement, why to use the full name “autoregressive distributed lag regression” again? Moreover, the sentence is not well written. Furthermore, the flow of these two sentences is not interesting.
Page 1, Lines 22-23: Authors’ statemen: “This empirical finding indicates that the rise of economic policy uncertainty in each country in the long term will decrease the sustainable stock market returns”. Suggestion: Please revise this statement. This study does not talk about forecasting of future trends; therefore, the authors should not say “will”. It should be “is found to decrease”, not “will decrease”.
Overall comments for abstract: The abstract is not well written. In addition to language mistakes, the flow, sequence, and organization is not good. It is also not clear what kind of impact the examine related to gold, bitcoin, and oil? The impact of their prices, volatility, or shocks on the selected stock markets? Similarly, I am not convinced with the policy suggestions given by the authors, “it is recommended that the policy makers in these 12 countries should focus on the stability of economic policy to reduce the negative effect of EPU on stock market returns in the long run”. This suggestion is certainly hypothetical, as policy makers or any government cannot control such elements, specifically in democratic government structure (the sample selected by the authors). It is obvious the authors have not put their sincere efforts in writing and proofreading the paper.
The introduction section similarly does not formulate a compelling story, and the connection among different parts of the introduction is weak. There is no discussion on the methodology employed and sample chosen. Therefore, the results and discussion are not rigorous.
Overall, I believe the manuscript in its current form is below acceptance standard required.
Author Response
Thank you very much for your invaluable comments and suggestions, which have improved the revised version significantly.
We would also like to send our appreciation to you for your time and efforts in reviewing our paper. We would also like to send our appreciation to you for your time and efforts in reviewing our paper and for providing the excellent comments. Below are our responses to your helpful comments and suggestions.
Question 1. Does the introduction provide sufficient background and include all relevant references? (must be improved)
Answer 1: Thank you very much for your advice. We have improved the introduction to provide sufficient background and include all relevant references in our revised manuscript.
Question 2. Are all the cited references relevant to the research? (must be improved)
Answer 2: Thank you very much for your advice. We have improved both our citations and the references so that all the cited references are relevant to the research of our paper of our revised manuscript.
Question 3. Is the research design appropriate? (can be improved)
Answer 3: Thank you very much for your advice. We have improved the research design and made it more appropriate in our revised manuscript.
Question 4. Are the methods adequately described? (must be improved)
Answer 4: Thank you very much for your advice. We have described the methods adequately in our revised manuscript.
Question 5. Are the results clearly presented? (must be improved)
Answer 5: Thank you very much for your advice. We have improved the presentation of our result to make it clearer in our revised manuscript.
Question 6: Are the conclusions supported by the results? (can be improved)
Answer 6: Thank you very much for your advice. We have improved the conclusions which are supported by the results in our revised manuscript.
Question 7: Moderate English changes required
Answer 7: Thank you very much for your advice. We have read and polished our paper carefully.
Question 8. Page 1, Lines 17-19: Following statements by the authors seem redundant in the abstract. “The quantitative research design will be used in this research. The data collection method uses secondary data obtained from the Thomson Reuters DataStream for stock data, economic policy uncertainty data from the EPU database and the World Bank database for macroeconomic data collection”. Suggestion: Please, improve the abstract. It is poorly written and not well organized.
Answer 8: Thank you very much for your advice. We have improved, written and organized the abstract better in our revised manuscript.
Question 9. Page 1, Lines 20-21: Authors’ statement “This study investigates the impact of EPU on stock market returns by using panel autoregressive distributed lag (ARDL) to examine the long-term cointegration and short-term effect”. Comment: Why only EPU? How about gold oil, and bitcoin? They are not judged using ARDL?
Answer 9: Thank you very much for your advice. We have explained why we only investigate the impact of EPU on stock market returns by using panel autoregressive distributed lag (ARDL) to examine the long-term cointegration and short-term effect clearly in our revised manuscript.
Question 10. Page 1, Lines 22-23: Authors’ statement “The results of the autoregressive distributed lag regression indicate that EPU has a negative and significant effect in the long term. While gold, oil and bitcoin have a positive and significant impact on the sustainable stock market returns”. Comment: As the abbreviation “ADRL” has been used in earlier statement, why to use the full name “autoregressive distributed lag regression” again? Moreover, the sentence is not well written. Furthermore, the flow of these two sentences is not interesting.
Answer 10: Thank you very much for your advice. We have used ADRL instead and rewritten the sentence in our revised manuscript.
Question 11. Page 1, Lines 22-23: Authors’ statemen: “This empirical finding indicates that the rise of economic policy uncertainty in each country in the long term will decrease the sustainable stock market returns”. Suggestion: Please revise this statement. This study does not talk about forecasting of future trends; therefore, the authors should not say “will”. It should be “is found to decrease”, not “will decrease”.
Answer 11: Thank you very much for your advice. We have revised the statement in our revised manuscript.
Question 12. Overall comments for abstract: The abstract is not well written. In addition to language mistakes, the flow, sequence, and organization is not good. It is also not clear what kind of impact the examine related to gold, bitcoin, and oil? The impact of their prices, volatility, or shocks on the selected stock markets? Similarly, I am not convinced with the policy suggestions given by the authors, “it is recommended that the policy makers in these 12 countries should focus on the stability of economic policy to reduce the negative effect of EPU on stock market returns in the long run”. This suggestion is certainly hypothetical, as policy makers or any government cannot control such elements, specifically in democratic government structure (the sample selected by the authors). It is obvious the authors have not put their sincere efforts in writing and proofreading the paper.
Answer 12: Thank you very much for your advice. We have rewritten the abstract and the entire paper, not only corrected all language mistakes, but also taken care of the flow, sequence, and organization in our revised manuscript. We have also improved the interpretation and policy inferences drawn from our results in our revised manuscript.
Question 13: The introduction section similarly does not formulate a compelling story, and the connection among different parts of the introduction is weak. There is no discussion on the methodology employed and sample chosen. Therefore, the results and discussion are not rigorous.
Answer 13: Thank you very much for your advice. We have improved the introduction section and made it formulate a compelling story, and connected better among different parts of the introduction in our revised manuscript. We have discussed the methodology employed and sample chosen in our paper and rewritten the results and discussion so that they are rigorous in our revised manuscript.
Question 14: Overall, I believe the manuscript in its current form is below acceptance standard required.
Answer 14: Thank you very much for your advice. We have improved the entire manuscript and we wish you will find the current form reaches required acceptance standard.
We hope that you will find this manuscript suitable to be included in an upcoming issue of your publication.

Reviewer 2 Report
1. topic is interesting and worth investigation for these countries.
2. Literature review is coherent but needs to be improved and updated .
3. the link between sections is balanced.
4. Models are well specified.
5. Result section needs to be expanded and include more information on findings.
5. Conclusion section is short comparing to the rest of paper. Author(s) need to include policy implication.
6. Paper needs to be professional English edited.
Author Response
Thank you very much for your invaluable comments and suggestions, which have improved the revised version significantly.
We would also like to send our appreciation to you for your time and efforts in reviewing our paper. We would like to thank you for your following comments:
- Are the methods adequately described? (yes)
- English language and style are fine/minor spell check required
- topic is interesting and worth investigation for these countries.
- Literature review is coherent .
- the link between sections is balanced.
- Models are well specified.
Below are our responses to your helpful comments and suggestions.
Question 1. Does the introduction provide sufficient background and include all relevant references? (can be improved)
Answer 1: Thank you very much for your advice. We have improved the introduction to provide sufficient background and include all relevant references in our revised manuscript.
Question 2. Are all the cited references relevant to the research? (can be improved)
Answer 2: Thank you very much for your advice. We have improved both our citations and the references so that all the cited references are relevant to the research of our revised manuscript.
Question 3. Is the research design appropriate? (can be improved)
Answer 3: Thank you very much for your advice. We have improved the research design and made it more appropriate in our revised manuscript.
Question 4: Are the conclusions supported by the results? (must be improved)
Answer 4: Thank you very much for your advice. We have improved the conclusions which are supported by the results in our revised manuscript.
Question 5: Literature review is coherent but needs to be improved and updated .
Answer 5: Thank you very much for your advice. We have improved and updated the literature review in our revised manuscript.
Question 6. Result section needs to be expanded and include more information on findings.
Answer 6: Thank you very much for your advice. We have expanded and included more information on the findings in our revised manuscript.
Question 7. Conclusion section is short comparing to the rest of paper. Author(s) need to include policy implication.
Answer 7: Thank you very much for your advice. We have included policy implications in the Conclusion section of our revised manuscript.
Question 8. Paper needs to be professional English edited.
Answer 8: Thank you very much for your advice. We have got a professional editor to polish our revised manuscript.
We hope that you will find this manuscript suitable to be included in an upcoming issue of your publication.

Reviewer 3 Report
From the overall presentation I would say that interesting research work has been done. The topic is also important for the readers of the journal. However, I have a few more significant challenges with the paper.
- You should include some hypotheses and test them.
- Explain why you chose to analyze twelve countries.
- The research methods used are appropriate but have limitations, and this should be mentioned. The validation of the models could be presented and justified. Furthermore, the uncertainties of the applied analysis could be discussed.
- The discussion and implications are rather short, and they should be extended.
- Conclusions should be expanded, pointing out the limitations of the paper.
Author Response
Thank you very much for your invaluable comments and suggestions, which have improved the revised version significantly.
We would also like to send our appreciation to you for your time and efforts in reviewing our paper. We would like to thank you for your following comments:
- English language and style are fine/minor spell check required
- From the overall presentation I would say that interesting research work has been done.
- The topic is also important for the readers of the journal.
- The research methods used are appropriate
Below are our responses to your helpful comments and suggestions.
Below are our responses to your helpful comments and suggestions.
Question 1. Does the introduction provide sufficient background and include all relevant references? (can be improved)
Answer 1: Thank you very much for your advice. We have improved the introduction to provide sufficient background and include all relevant references in our revised manuscript.
Question 2. Are all the cited references relevant to the research? (can be improved)
Answer 2: Thank you very much for your advice. We have improved both our citations and the references so that all the cited references are relevant to the research of our paper of our revised manuscript.
Question 3. Is the research design appropriate? (can be improved)
Answer 3: Thank you very much for your advice. We have improved the research design and made it more appropriate in our revised manuscript.
Question 4. Are the methods adequately described? (can be improved)
Answer 4: Thank you very much for your advice. We have described the methods adequately in our revised manuscript.
Question 5. Are the results clearly presented? (can be improved)
Answer 5: Thank you very much for your advice. We have improved the presentation of our result to make it clearer in our revised manuscript.
Question 6: Are the conclusions supported by the results? (can be improved)
Answer 6: Thank you very much for your advice. We have improved the conclusions which are supported by the results in our revised manuscript.
Question 6: You should include some hypotheses and test them.
Answer 6: Thank you very much for your advice. We have included some hypotheses and test them in our revised manuscript.
Question 7. Explain why you chose to analyze twelve countries.
Answer 7: Thank you very much for your advice. We have explained why we choose to analyze twelve countries in our revised manuscript.
Question 8. The research methods used are appropriate but have limitations, and this should be mentioned. The validation of the models could be presented and justified. Furthermore, the uncertainties of the applied analysis could be discussed.
Answer 8: Thank you very much for your advice. We have improved the research methods, included the validation of the models, and discussed the uncertainties of the applied analysis in our revised manuscript.
Question 9: The discussion and implications are rather short, and they should be extended.
Answer 9: Thank you very much for your advice. We have extended the discussion and implications of our revised manuscript.
Question 10: Conclusions should be expanded, pointing out the limitations of the paper.
Answer 10: Thank you very much for your advice. We have expanded the conclusions and pointed out the limitations of the paper in our revised manuscript.
We hope that you will find this manuscript suitable to be included in an upcoming issue of your publication.

Round 2
Reviewer 1 Report
The manuscript is now improved and the flow of the work done is better. I will suggest an interesting future extension of this work to be included in the Conclusion section of the paper: that is, examining four-moment modified value at risk and conditional value at risk (using Cornish-Fisher Expansion) of different mixed portfolios pairing the assets under study (equity-oil and equity-gold portfolios). Please add this as the future extension of the manuscript under study and give credit to the relevant work by Ali, Jiang, and Sensoy (2021).
Reference: Ali, F., Jiang, Y., and Sensoy, A. (2021). Downside risk in Dow Jones Islamic equity indices: Precious metals and portfolio diversification before and after the COVID-19 bear market. Research in International Business and Finance, Volume 58, 101502. DOI: https://doi.org/10.1016/j.ribaf.2021.101502.
Good luck.